Marked increase in rat red blood cell membrane protein glycosylation by one-month treatment with a cafeteria diet

Oliva Laia 1
Baron Cristian 1
Fernández-López José-Antonio 1 2 3
Remesar Xavier 1 2 3
Alemany Marià 1 2 3 malemany@ub.edu
1 Department of Nutrition and Food Science, Faculty of Biology, University of Barcelona , Barcelona , Spain
2 Institute of Biomedicine of the University of Barcelona , Barcelona , Spain
3 CIBER OBN , Barcelona , Spain
Foti Daniela
Electronic publication date: 2015 Jul 16
Publication date: 2015
Volume: 3
Electronic Location ID: e1101
Received 2015 Mar 27; Accepted 2015 Jun 22
Copyright: © 2015 Oliva et al.
Copyright year: 2015
Copyright holder: Oliva et al.
License: This is an open access article distributed under the terms of the Creative Commons Attribution License, which permits unrestricted use, distribution, reproduction and adaptation in any medium and for any purpose provided that it is properly attributed. For attribution, the original author(s), title, publication source (PeerJ) and either DOI or URL of the article must be cited.
License URL: https://creativecommons.org/licenses/by/4.0/

Keywords: Erythrocyte, Protein glycosylation, Glycemia, Glycosylated hemoglobin, Blood cell membrane, Cafeteria diet, Glycosylation

Funding: Plan Nacional de Investigación en Biomedicina SAF2012-34895 Plan Nacional de Ciencia y Tecnología de los Alimentos AGL-2011-23635 CIBER OBN This study was done with the partial support of grants of the Plan Nacional de Investigación en Biomedicina (SAF2012-34895) and the Plan Nacional de Ciencia y Tecnología de los Alimentos (AGL-2011-23635) of the Government of Spain, and assistance from the CIBER OBN. The funders had no role in study design, data collection and analysis, decision to publish, or preparation of the manuscript.

==============================
Background and Objectives. Glucose, an aldose, spontaneously reacts with protein amino acids yielding glycosylated proteins. The compounds may reorganize to produce advanced glycosylation products, which regulatory importance is increasingly being recognized. Protein glycosylation is produced without the direct intervention of enzymes and results in the loss of function. Glycosylated plasma albumin, and glycosylated haemoglobin are currently used as index of mean plasma glucose levels, since higher glucose availability results in higher glycosylation rates. In this study we intended to detect the early changes in blood protein glycosylation elicited by an obesogenic diet.

Experimental Design. Since albumin is in constant direct contact with plasma glucose, as are the red blood cell (RBC) membranes, we analyzed their degree or glycosylation in female and male rats, either fed a standard diet or subjected to a hyper-energetic self-selected cafeteria diet for 30 days. This model produces a small increase in basal glycaemia and a significant increase in body fat, leaving the animals in the initial stages of development of metabolic syndrome. We also measured the degree of glycosylation of hemoglobin, and the concentration of glucose in contact with this protein, that within the RBC. Glycosylation was measured by colorimetric estimation of the hydroxymethylfurfural liberated from glycosyl residues by incubation with oxalate.

Results. Plasma glucose was higher in cafeteria diet and in male rats, both independent effects. However, there were no significant differences induced by sex or diet in either hemoglobin or plasma proteins. Purified RBC membranes showed a marked effect of diet: higher glycosylation in cafeteria rats, which was more marked in females (not in controls). In any case, the number of glycosyl residues per molecule were higher in hemoglobin than in plasma proteins (after correction for molecular weight). The detected levels of glucose in RBC were lower than those of plasma, even when expressed in molal units, and were practically nil in cafeteria-diet fed rats compared with controls; there was no effect of sex.

Conclusions. RBC membrane glycosylation is a sensitive indicator of developing metabolic syndrome-related hyperglycemia, more sensitive than the general measurement of plasma or RBC protein glycosylation. The extensive glycosylation of blood proteins does not seem to be markedly affected by sex; and could be hardly justified from an assumedly sustained plasma hyperglycemia. The low levels of glucose found within RBC, especially in rats under the cafeteria diet, could hardly justify the extensive glycosylation of hemoglobin and the lack of differences with controls, which contained sizeable levels of intracellular glucose. Additional studies are needed to study the dynamics of glucose in vivo in the RBC to understand how such extensive protein glycosylation could take place.

Introduction

Glucose, in addition to being the main intercellular energy staple, is a reducing aldose. Thus, it may easily react with a number of chemical groups in proteins and other biological compounds. The direct condensation with protein free amino groups (Maillard reactions) (John & Lamb, 1993) is fairly common, to the degree that a significant proportion of circulating plasma proteins are glycosylated (Gragnoli et al., 1982), as well as proteins in the red blood cell (RBC) membrane (Miller, Gravallese & Bunn, 1980) and the hemoglobin they contain (Bunn, Gabbay & Gallop, 1978). The proportion of hemoglobin glycosylated in the terminal valine of chain B (Hb1AC) is currently used as an index of overall exposure to free plasma glucose over time (Siu & Yuen, 2014). Glycosylation products may undergo Amadori reorganizations, producing a number of complex compounds known as advanced glycosylation products (AGP) (Henning et al., 2011), which play a significant role in the control of substrate utilization (Wu et al., 2011), cell function (Guo et al., 2012) and inflammation (Poulsen et al., 2014).

The chemical reactivity of glucose is often overlooked because of its overwhelming function in energy supply and rapid turnover, but direct non-enzymatic glycosylation remains a common mechanism of alteration of protein function and interference in signaling pathways (Asahi et al., 2000; Itkonen & Mills, 2013). It is commonly accepted that higher sustained circulating levels of glucose, as in diabetes, result in increased proportions of glycosylated proteins in plasma, RBCs and endothelial cells, Hb1AC being a case in point (Carson et al., 2010). In fact, equations based on the correlation between mean estimated plasma glucose concentration and Hb1AC proportion are currently in use (Borg & Kuenen, 2009; Nathan et al., 2008).

The self-selected cafeteria diets (Sclafani & Springer, 1976) are essentially hyperlipidic (Prats et al., 1989), and its consumption by rats causes hyperphagia, insulin resistance and obesity (Correa Pinto & Monteiro Seraphim, 2012; Prats et al., 1989). Exposure for one month of young adult rats to a cafeteria diet induces a number of metabolic changes that are in the limit of normalcy and correspond to the initial stages of the metabolic syndrome (Romero et al., 2010). The effects are more marked in male than in female rats (Romero et al., 2012), probably because of the anti-inflammatory effects of estrogen (Thomas et al., 2003); but, in any case, the obesity is already patent. Short-term treatment with cafeteria diets induce a mild hyperglycemia and hyperinsulinemia (Romero et al., 2010), but not frank diabetes, which is more developed after prolonged exposure (Castell-Auví et al., 2012; Correa Pinto & Monteiro Seraphim, 2012).

In the present study, we have analyzed whether the glycosylation degree of total plasma or RBC proteins, as well as those of RBC membranes, are a direct correlate of their prolonged contact with plasma glucose in an early stage of development of hyperglycemia. We wanted, also, to check whether sex exerts any influence on the glycosylation response to comparable glucose concentrations.

Materials and Methods

Animals and animal handling

All animal handling procedures were carried out in accordance with the norms of the European, Spanish and Catalan Governments. The study was specifically approved (DMAH-5483) by the Animal Ethics Committee of the University of Barcelona.

Wistar adult male and female (9 week-old) rats were used (Harlan Laboratories Models, Sant Feliu de Codines, Spain). The rats were adapted to the Animal House environment for at least one week prior to the beginning of the experiment, and were fed a standard (Harlan, type 2014) chow. The rats were kept in solid-bottomed adjoining collective 2-rat cages, with wood shards as bedding material. Half of the rats in each group were subjected to an energy-rich limited-item cafeteria diet (Ferrer-Lorente et al., 2005) for a month. The items of cafeteria diet (plain cookies spread with liver pâté, bacon, standard chow, water and whole cow’s milk containing 300 g/L sucrose and a mineral and vitamin supplement) were renewed daily. Food consumption per cage and rat weights were recorded every day.

The four experimental groups (N = 6 for each) were: female-control (FC), female-cafeteria, (FCAF) male-control (MC) and male-cafeteria (MCAF). On day 29, a small sample of blood was taken from a cut in the rat tail’s tip, centrifuged in capillary tubes, and the plasma was frozen for later measurement of glucose levels.

At the end of the experiment (day 30), the rats were anaesthetized with isoflurane and killed by exsanguination (blood drawn from the aorta using a dry-heparinized syringe). Part of the blood was centrifuged immediately (at 1,300 × g for 25 min and 2–4 °C). Plasma and packed cells were frozen and kept at −20 °C. A sample of fresh blood was deproteinized with 0.5 volumes of 6,7 M perchloric acid, mixed, neutralized with 4.5 M KOH containing 1.55 M potassium bicarbonate, centrifuged again at the same speed (at 4 °C), and the supernatants used for the measurement of total blood glucose.

Packed cell volume was estimated from the weight of blood before centrifugation, that of plasma obtained after that centrifugation and the (redundant) weight of packed cells sedimented. Since the densities of cells and packed cells were known, and the proportion of packed cell volume was a direct correlate of time and acceleration generated during centrifugation, we used the previously described graphs, obtained under the same conditions (Romero et al., 2012) to estimate the actual proportion of plasma trapped between the cells, and thus determine the real packed cell volume.

A known weight of frozen packed cells was suspended in 10 volumes of chilled pure water. After gentle shaking for 20 min at 4 °C, the suspension was centrifuged 10 min at 2,000 × g and 2–4 °C. The clear supernatant (hemoglobin and cytosolic RBC proteins) was used for the analyses of total and glycosylated protein.

RBC membrane separation

About 0.5 g samples of frozen packed cells were weighed and suspended in 3.5 mL of chilled tris–HCl buffer 10 mM pH 7.4, the cells were gently stirred until a uniform solution was obtained. Then, 4 mL of chilled 250 mM glucose were added and gently mixed. After standing 15 min (Tomoda et al., 1984), the suspension was coarse-filtered through a small wad of glass fiber to remove debris, and then was centrifuged for 3 min at 8,000 × g in the cold (2–4 °C). The fluffy precipitate was suspended in medium, and centrifuged again. A small translucent sediment of RBC membranes was obtained; it was weighed, and used for the analysis of protein, total phosphate and glycosylation.

Chemical analyses

Glucose in plasma and deproteinized fresh blood was measured with a glucose oxidase kit (Biosystems, Barcelona, Spain), supplemented with mutarotase (490 nkat/mL of reagent) (Calzyme, San Luis Obispo, CA, USA). Mutarotase was added to speed up epimerization equilibrium of α- and β-D-glucose and thus facilitate the oxidation of β-D-glucose by glucose oxidase (Miwa et al., 1972). The enzyme addition was complemented with a precise control of the time (15 min) and temperature (30 °C) conditions of development of the reaction, in order to make sure all glucose in the sample was oxidized to gluconate. Protein content was estimated with a variant of the Lowry method (Lowry et al., 1951) using fatty acid-free bovine serum albumin (Sigma, St Louis MO, USA) as standard.

RBC membranes were mineralized with perchloric acid (700 g/L) in 15 mL Teflon-stoppered glass tubes, in a dry block heater, at 150 °C for 24 h (Stein & Smith, 1982). Aliquots of the clear mineralized samples were used, after centrifugation, for the estimation of phosphate using the phosphomolybdate reaction using sodium mono-phosphate as standard (Gomori, 1942; Stein & Smith, 1982). A standard of phosphatidyl-choline (Sigma) was processed along with the samples. The measurements of phosphate from the phosphatidyl-choline standards proved that mineralization was complete (98–101%). Each batch of samples was corrected using their own standards, ran in parallel.

The degree of glycosylation was estimated by direct measurement of the 5-hydroxymethylfurfural (HMF) liberated by treatment of the samples with 1 N oxalic acid at 100 °C for 24 h (Gabbay et al., 1979) in 15 mL Teflon-stoppered tubes set in a dry heating block. After cooling, trichloroacetic acid was added (final concentration 100 g/L), and the tubes were shaken and centrifuged for 15 min at 5,000 × g. The precipitate was discarded. The amount of HMF released was measured through the condensation of HMF with 50 mM thiobarbituric acid (Sigma) (Gabbay et al., 1979). After 20 min at 37 °C for development of color, the OD was measured at 443 nm, using blanks and pure HMF (Sigma) standards, and was used to determine the HMF (i.e., unaltered glycosyl residues in proteins) in each sample.

Blood cell glucose estimation

Blood glucose is the composite of the glucose carried by the cells and that in plasma using the common (Higgins, Garlick & Bunn, 1982) formula: blood glucose=plasma glucose ×1−PCV+cell glucose ×PCV

where PCV (Packed Cell Volume) is the net cell volume fraction (i.e., discounting trapped plasma volume) of total blood volume (in this case =1). In that equation, we had, for each rat, the PCV value as well as plasma and blood glucose. Crude cell-transported glucose was derived from these data. Since it was assumed that trapped plasma glucose concentration was the same than in plasma obtained by centrifugation, the glucose present in that plasma fraction was discounted from the total packed cell glucose (and added to the final data for “plasma glucose”). These calculations were carried out for each individual rat, thus all data used for the calculations were homologous.

Statistics

Statistic comparison between groups was carried out using 2- and 3-way ANOVA analyses, and the Bonferroni post-hoc test for further differences between specific groups (Prism 5 program; GraphPad Software, La Jolla, CA, USA).

Results

Table 1 presents the changes in body weight experienced by the rats during one month of exposure to a cafeteria diet. The initial weight difference between female and male rats widened with time, since control males increased about 20% of their weight, compared with 15% of females; cafeteria diet increased body weight 35% in males and 34% in females. Males ate more energy than females: 36% (control diet) or 19% (cafeteria diet). Males’ food (expressed as energy content) intake was 2.5-fold higher in cafeteria than in control diet; the value for females was 2.8×.

Table 1 Body weight changes, energy intake and plasma glucose of Wistar rats fed control or cafeteria diet for 30 days.

The data are the mean ± sem of 6 animals per group. Plasma glucose was measured on day 29. Statistical significance of the differences between means were determined using a 2-way ANOVA program.

Parameter	Units	Male	Female	P values	
		Control	Cafeteria	Control	Cafeteria	Sex	Diet	Interaction	
Initial weight	g	394 ± 9	379 ± 3	238 ± 5	217 ± 4	<0.0001	NS	NS	
Final weight	g	474 ± 10	511 ± 3.5	275 ± 11	290 ± 8	0.0068	<0.0001	NS	
Weight increase	g/30 d	82 ± 10	137 ± 4	41 ± 5	74 ± 7	<0.0001	<0.0001	NS	
Eenergy intake	MJ/30 d	8.62 ± 0.04	21.4 ± 1.5	6.32 ± 0.39	18.0 ± 1.0	0.0055	<0.0001	NS	
W	3.33 ± 0.01	8.26 ± 0.59	2.44 ± 0.15	6.93 ± 0.38		
Plasma glucose	mM	7.58 ± 0.32	9.13 ± 0.15	6.83 ± 0.26	8.53 ± 0.12	0.0082	<0.0001	NS	

Standard plasma glucose (measured on day 29) showed both an effect of sex (female levels being lower) and diet (cafeteria diet data being higher).

Table 2 shows the data obtained from the analysis of blood extracted under isoflurane anesthesia. In this case, all plasma glucose data were higher than those obtained on day 29 under basal conditions, and there were no statistical differences between the groups attributable to sex or diet. Total blood glucose values were lower than those of plasma, and showed neither differences by sex or diet. However, the estimated data for cell glucose showed a clear effect of diet (Fig. 1). In both groups of cafeteria rats, the levels were minimal, not statistically different from zero, while those of rats under the control diet were lower than in plasma but clearly positive, the differences being not significant for “sex” but significant for “diet”. In control rats, when water content of plasma (about 92%) and packed cells (about 70%) was included in the calculations, the molal concentrations of cell glucose were in the range of 1/3rd of those of plasma; female rats presented similar values. Cafeteria diet-fed rats showed values in the range of only 4–7%.

Figure 1 Distribution of blood glucose in plasma and cell compartments of Wistar rats fed control or cafeteria diet during 30 days.

Data are the mean ± sem of 6 animals per group (killed under isoflurane anesthesia), and were calculated from the data presented in Table 2. Statistical significance (two-way ANOVA) of the differences between groups: No differences were found for “sex”, but “diet” showed P < 0.0001 for cells and was not significant for plasma. Blood cell data for cafeteria diet were not statistically different from zero.

Table 2 Blood glucose and packed cell volume of Wistar rats fed control or cafeteria diet for one month, after exsanguination under isoflurane anesthesia on day 30.

The data are the mean ± sem of 6 animals per group. Packed cell volume data were corrected for trapped plasma as explained in the text. Statistical significance of the differences between means were determined using a 2-way ANOVA program.

Parameter	Units	Male	Female	P values	
		Control	Cafeteria	Control	Cafeteria	Sex	Diet	Interaction	
Blood glucose	mM	6.83 ± 0.13	5.93 ± 0.37	6.43 ± 0.14	6.31 ± 0.31	NS	NS	NS	
Plasma glucose	mM	10.41 ± 0.33	10.90 ± 0.64	10.71 ± 0.63	10.84 ± 0.64	NS	NS	NS	
Packed cell volume	% blood volume	45.7 ± 0.9	43.8 ± 0.7	43.1 ± 1.1	42.7 ± 1.8	NS	NS	NS	
Blood cell glucosea	µmol/g	2.75 ± 0.32	0.53 ± 0.96*	2.76 ± 0.56	0.33 ± 0.27*	NS	<0.0001	NS	
Notes.

a Blood cells’ glucose concentration was calculated for each animal from glucose data (whole blood and plasma) and the net packed cell volume.

* Statistically not different from zero.

The proportions of glycosylated protein, both in RBC and in plasma, are presented in Fig. 2. No significant differences were observed between the groups for “sex” and “diet”. However, cell protein was more heavily glycosylated than plasma proteins. In the case of cells, since most of the protein (>95%) is hemoglobin, it can be assumed that most glycosyl residues were bound to this protein; since its molecular weight (tetramer) is about 64,000, the molar ratio of HMF to hemoglobin was about 320, i.e., about 80 glycosyl residues per hemoglobin subunit. This value is about six-fold higher than the number of sites representing 7% Hb1AC, which is limited to the terminal chains of hemoglobin. In the case of plasma, since albumin makes about 55% of plasma proteins and its molecular weight is close to 66,500, we obtain about 90 glycosyl residues per molecule. Evidently, this is only an imprecise approximation but shows that under the particular conditions of this experiment, protein glycosylation was significant and about 3.5 times more intensive in cells (on a molar ratio) than in plasma proteins as a whole.

Figure 2 Degree of glycosylation expressed in nmol HMF per mg total protein in the cells and plasma of Wistar rats fed control or cafeteria diet during 30 days.

Data are the mean ± sem of 6 animals per group. Statistical significance (three-way ANOVA) of the differences between groups: No differences were found for “sex” and “diet”, but the differences between “compartments” (i.e., blood cells vs. plasma) was P < 0.0001.

Figure 3 depicts the rate of glycosylation observed in purified membranes of blood cells. Since purification of membranes is not even close to quantitative, we could not determine in which proportion RBC membranes were glycosylated. In fact, we were not able to ascertain the degree of the purity of samples. Thus, membrane proteins could be contaminated by hemoglobin (in spite of the appearance of total elimination at the expense of dwindling recovery of membranes), spectrin or other molecules. Thus, we decided to also relate the degree of glycosylation to phospholipid, an exclusive membrane component in RBC. The molar ratio of released HMF to phospholipid phosphate (Fig. 3) showed an image quite different from that of Fig. 2. There were no statistical differences between groups attributable to “sex”. This was clear for control diet, but the post-hoc test showed a significant (P < 0.05) sex-related difference in cafeteria-fed rats. The effect of “diet” was significant, with several-fold higher values in cafeteria- than in control-fed rats. Presentation of the data of HMF per mg of membrane preparation protein shown in Fig. 3, yields almost the same pattern, but statistical significance was lower because the individual variation of data was higher.

Figure 3 Degree of glycosylation of blood cell membranes, expressed as mmoles of HMF per mmol of phospholipid P or unit of membrane protein weight, of Wistar rats fed control or cafeteria diet during 30 days.

Data are the mean ± sem of 6 animals per group. Statistical significance (two-way ANOVA) of the differences between groups: When analyzed for HMF vs. protein, no statistical differences were found. When analyzed for HMF vs. phospholipid P. No differences were found for “sex” but the difference for “diet” was P < 0.0001. There was a significant interaction between sex and diet. Glycosylation was higher (P < 0.05, Bonferroni post-hoc test) in the female cafeteria rats, compared with males.

Discussion

In the development of this apparently simple study, we tried to maintain a close control of methodology, since the problems of glycosylation of blood components have generated a sizeable number of studies, but their integrated analysis is scarce, in a way that only limited comparisons have been studied. We intended to present homologous data for plasma and RBC proteins, including also samples of RBC membranes, and using a model in which the metabolic syndrome pathologies, especially insulin resistance and hyperglycemia, were not fully set in.

The problem of anesthesia as hyperglycemic agent (Arola et al., 1981; Zuurbier et al., 2008) has not been solved; we opted by using this avenue to obtain sufficient blood to carry out all the compartmentation and membrane experiments in the same samples. Consequently we had to obtain separate plasma samples to compare the basal results with previous studies (Palou et al., 1980). The changes elicited by cafeteria diet agree with previously published studies (Ferrer-Lorente et al., 2005). We assumed that the brief isoflurane anesthesia-induced hyperglycemia (Zuurbier et al., 2008) (less than 5 min from start to exsanguination) changes plasma glucose levels, but its effects on RBC glucose (if any) would be at least partly buffered. In any case, it is highly improbable that these changes would affect differentially the rats depending on their diet. The uniformity of the data obtained seem to support this weak point of our experimental setup. We have not been able to circumvent the problem within the ethical standards of our Laboratory.

The lower blood vs. plasma glucose levels, more marked in cafeteria diet-fed rats, attest directly to a lower cell compartment glucose content (there were minimal differences in packed cell volume). The accuracy of the calculations used to quantify the cell glucose content notwithstanding, do not change the fact that cafeteria rats had higher glucose content in the blood plasma fraction compared with that of cells; precisely the glucose in direct contact with hemoglobin.

Metabolic syndrome, diabetes and in general, high exposure to inflammation and hyperglycemia increase the glycosylation of plasma proteins (Matsuura, Hughes & Khamashta, 2008; Roohk & Zaidi, 2008). In fact, glycosylated albumin has been proposed as an indicator of maintained hyperglycemia (i.e., exposure of plasma proteins to higher aldose levels for long periods) (Abe & Matsumoto, 2008). However, the most used indicator of long-time maintained hyperglycemia is the measurement of glycosylated hemoglobin (Siu & Yuen, 2014), which initially was applied to whole RBC hemoglobin (Carson et al., 2010), but soon was focused on the terminal amino groups of hemoglobin (Hb1AC) alone, giving rise to a much more sensitive (and extended) assay methodology (Little et al., 2008; Weykamp et al., 2008). The study of glycosylated hemoglobin (Hb1AC) has become one of the standard elements for the evaluation of diabetes (metabolic syndrome) and, in general, sustained hyperglycemia (Ong et al., 2010). The critical point, however, is that all hemoglobin is contained within the RBC, and is not in direct contact with plasma glucose. This obvious circumstance would make a priori glycosylated albumin a more acceptable indicator of hyperglycemia. The conundrum of a marker of glycosylation not in direct contact with the parameter it measures has not been sufficiently explained so far. Nevertheless, its widespread use and clinical reliability are powerful reasons in favor of its continued use despite the largely unexplained nature of its origins.

In mammals, the direct permeability of the RBC membrane to glucose is low, if any (Britton, 1964; Rich et al., 1967), however, interchange of plasma and RBC glucose is active thanks to a facilitated-diffusion transport system (Levine, Oxender & Stein, 1965). The transport has been attributed, mainly to GLUT1 (Graybill et al., 2006), which function may be regulated by insulin, glucocorticoids and other factors (Kahn & Flyer, 1990). However, no differences in glucose transport through erythrocyte membranes were found between diabetic and euglycemic children (Mortensen & Brahm, 1985). There is, also, a high variability in the permeability of RBC membranes to glucose, due to species differences, individual factors and transporter modulation/saturation (Khera et al., 2008).

Compartmentation of blood glucose between plasma and cells may be an important regulatory factor by itself (Palou et al., 1980), since glucose carried by blood cells is rapidly interchanged with tissues (Jacquez, 1984). This is in overt contradiction with the slow velocity of glucose interchange of RBC when measured in vitro (Sen & Widdas, 1962). In addition, given the glycolytic nature of mammalian RBC, it can be expected that a sizeable part of the glucose entering the cell is rapidly glycolyzed to lactate, a process that is the only significant source of ATP for the cell. This inefficient mechanism converts blood in a sizeable source of lactate, which implies that a variable part of glucose will be converted to hexoses-P on entering the cell, and thus (at least in part, when isomerized to ketose-P) lose its aldose-related glycosylating capacity.

The high proportions of Hb1AC found under conditions of assumed sustained hyperglycemia (Giuffrida et al., 2010) contrast with the physical existence of barriers between hyperglycemic plasma and hemoglobin. We could not explain why Hb1AC is so highly correlated with hyperglycemia, since total hemoglobin glycosylation does not reflect only hyperglycemia (Adams et al., 2009; Chan et al., 2014), which is in agreement with our data. In addition, diabetogenic conditions, such as those presented here and those found in the literature (Koga et al., 2007; Miyashita et al., 2007; Wakabayashi, 2012) do not show the expected changes in hemoglobin glycosylation. Our data on the lack of significant changes elicited by diet on plasma protein glycosylation do not agree with the common occurrence of increased glycosylated proteins in plasma of humans and rodents alike under already settled metabolic syndrome or its associated pathologies (Gornik & Lauc, 2008). The probable differences lie in the fact that in our model of initial stages of metabolic syndrome, the pathologic markers have not been yet developed fully, as we have previously found (Ferrer-Lorente et al., 2005). It must be also taken into account that metabolic syndrome-induced modifications on plasma proteins (Marliss et al., 2006; Welle et al., 1992) and RBC (Cohen, Franco & Joiner, 2004; Manodori & Kuypers, 2002) increase their cell turnover rates which compounds the problem and makes more difficult the comparisons unless the data maintain their homology.

The elevated degree of glycosylation found in RBC membranes, however, shows that even the small differences in basal glycemia found in our model are enough to already induce several-fold changes in the glycosylative activity of plasma glucose. Probably, other factors so far not identified, may help explain the increased glycosylation observed even at early stages of the development of metabolic syndrome. The relationship with high-energy (lipid) diet is clear, but the common assumption that these changes are a correlate of hyperglycemia remain unproven, and largely based only on indirect evidence.

In our experiment, the degree of glycosylation of hemoglobin was high, even under conditions in which practically no free glucose was found within the RBC. We only measured glycosyl residues, not those recombined by Amadori rearrangements, i.e., those able to liberate hydroxymethylfurfural. However, the ratio of up to 90 glycosyl residues per subunit of hemoglobin is close to the level of saturation of glycosylable sites. This was necessarily achieved in at most one month, a time shorter than the mean rat RBC half-life, 45–50 days (Burwell, Brickley & Finch, 1953), a value considerably decreased in rats with metabolic syndrome (Kung, Tseng & Wang, 2009). However, compared with albumin which median life span is close to 2 days (Reed et al., 1988), the differences can be better explained, since exposure of hemoglobin was 15-fold higher (30 days out of 45–50) than that of plasma proteins (2 days, assuming a behavior comparable to that of albumin). The shorter exposure was predictably more intense (as that of RBC membrane protein) because plasma proteins were in constant contact with plasma glucose.

The levels of glycosylation of plasma proteins and hemoglobin observed do not reflect the (limited) changes in plasma glucose, however, RBC membranes do. The results we obtained are puzzling; they agree with the known fact that exposure to hyperglycemia results in increased protein glycosylation, as shown by membrane proteins’ differences, but not observed in plasma proteins; this may be due to their shorter half-lives and limited span of glucose level change.

On the other side, the low free glucose levels observed in cafeteria diet-fed rat RBC, agree with a slower rate of uptake (Prats et al., 1987) compared with plasma, but cannot directly explain how the overall glycosylation of hemoglobin was unaffected by one month of consumption of a hyper-energetic obesogenic diet.

Conclusions

We conclude that blood glucose compartmentation, as previously indicated, may play a role, in the regulation of plasma/blood versus tissue glucose transport/transfer, more important than usually assumed, but also, that glycosylation of blood proteins widely affects non-diabetic young experimental animals, both under standard or hyper-energetic diet conditions. This extensive glycosylation does not seem to be markedly affected by sex; and could be hardly justified from an assumedly sustained plasma hyperglycemia. More detailed—and comprehensive—analyses should be carried out to study the dynamics of glucose in vivo in the RBC to understand how so extensive protein glycosylation as that found here could take place, including an special emphasis on the hormonal regulation of RBC glucose transporters.

We have also found that RBC membrane glycosylation is a sensitive indicator of developing metabolic syndrome-related hyperglycemia, more sensitive than the general measurement of plasma or RBC protein glycosylation.

Supplemental Information

File S1 Glucose. Contains all data on plasma, blood glucose and calculations for hematocrit and cell glucose

Click here for additional data file.

File S2 CellProtein. Contains the dfata for cell protein and HMF content

Click here for additional data file.

File S3 MembranePLandHMF. Contains the data on RBC membrane protein, phosphate from phospholipids and membrane HMF

Click here for additional data file.

File S4 Weights&Intakes: Contains the data on body weight changes and food intake

Click here for additional data file.

At the time this investigation took place, Christian Baron was an undergraduate student. Laia Oliva worked pro bono.

Additional Information and Declarations

Competing Interests

Author Contributions

Animal Ethics

The authors declare there are no competing interests.

Laia Oliva performed the experiments, prepared figures and/or tables, reviewed drafts of the paper.

Cristian Baron performed the experiments.

José-Antonio Fernández-López analyzed the data, contributed reagents/materials/analysis tools, prepared figures and/or tables, reviewed drafts of the paper.

Xavier Remesar analyzed the data, contributed reagents/materials/analysis tools, reviewed drafts of the paper.

Marià Alemany conceived and designed the experiments, analyzed the data, contributed reagents/materials/analysis tools, wrote the paper, prepared figures and/or tables.

The following information was supplied relating to ethical approvals (i.e., approving body and any reference numbers):

All animal handling procedures were carried out in accordance with the norms of European, Spanish and Catalan Governments. The study was specifically approved (DMAH-5483) by the Animal Ethics Committee of the University of Barcelona.

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
