# Peer review of "Marked increase in rat red blood cell membrane protein glycosylation by one-month treatment with a cafeteria diet"

_PeerJ, doi:10.7717/peerj.1101_

## Round 0.1 · original submission · Major Revisions

Many methodological aspects should be addressed, including experiment replication, glycemic variability, normalization of the glycation plasma and RBC products. Also, the counterintuitive finding of blood cell glucose reduction following diet should be discussed.

Reviewer 1 ·

Basic reporting

The article is clearly written. Only a few typing errors should be revised. E.g. in the abstract change “inn” for “in” and remove last bracket in the last sentence of the abstract. The article is properly put into context, with enough references. Most figures are relevant and well explained, although in my opinion I would remove Figure 1 since the information that it contains is redundant with that of table 1.

Experimental design

The manuscript describes original research in an interesting field. The experimental design is appropriate. There are some concerns about the methodology, but the authors address it properly in the discussion. Methods are detailed, although in my opinion a brief description of the cafeteria diet used would be of interest. If metabolic cages were used to quantify ingested food it should be stated. Also, there is an allusion to unpublished results in the materials section (line 101) that is not clear what it refers to. Ethics issues have properly been addressed.

Validity of the findings

The design contemplates enough replicates and statistics have been addressed to rely on the data. I recommend adding statistics concerning the increase in body weight that is mentioned in the first paragraph of the results section. I also suggest removing lines 170-172 in which authors mention data not shown that is normalized in a way that they have just discarded as reliable enough. In the discussion section, the lines 227-229 should be better explained since the meaning is not clear. Conclusions are clearly stated.

Reviewer 2 ·

Basic reporting

see below

Experimental design

see below

Validity of the findings

see below

Additional comments

This manuscript is directed towards comparing differences by gender and by diet in relative glycosylation of plasma proteins, hemoglobin and red cell membrane components in rats as a means of assessing changes with an animal model of metabolic syndrome. The manuscript would come across as more focused if the background and objectives portion of the abstract contained objectives.

I have a number of concerns about this manuscript. Essentially all the data are from a single experiment with no evidence the results were reproduced. This is probably most important with the estimate of marked reduction in blood cell glucose with diet, one of the more dramatic findings. The result doesn’t make much sense physiologically, hasn’t been reproduced, and depends on a subtraction methodology full of assumptions and for which there are considerable pitfalls. The membrane glycosylation is expressed per unit phospholipid phosphorus rather than per unit protein because the authors are concerned that they can’t distinguish what protein is membrane and what protein is hemoglobin contaminant. However if that is a concern how can they depend on the fraction of the HMF generated being from membrane proteins and not from contaminating glycated hemoglobin?

The ratios of glycation residue per hemoglobin and per albumin molecule do not sound plausible. It would be well for the authors to review relevant literature and see how consistent those findings are with previous reports. It doesn’t sound like necessary positive and negative controls are reported with those determinations to support the validity of the quantitation.

The authors do not adequately explain why there are such marked dIfferences in glucoses between tail vein blood and blood obtained by exsanguination. There are some comments about effects of the anesthetic to raise blood glucose but they are phrased in a fashion that is not transparent to a reader who was not a participant in the work. An inadequate explanation of those differences tends to cast doubt on the validity of other findings obtained on those samples.

The authors allude to differences in level of glycation per unit protein between plasma proteins and RBC proteins without conveying that they understand why those are and they express surprise at the greater extent of glycation of intra-RBC proteins than of plasma proteins. There are two key concepts commonly invoked for those findings which the authors don’t convey awareness of: one is differences among animal species in the permeability of the RBC membrane to glucose and the other is the dependence of accumulation of glycated residues on the turnover time of the underlying proteins. Hb is present in the circulation for a much longer interval than albumin. There is also some evidence on correlations correlations between level of glycosylation in plasma vs in red blood cell proteins which may be of interest:

Higgins PJ, Garlick RL, Bunn HF. Glycosylated hemoglobin in human and animal red cells. Role of glucose permeability. Diabetes. 1982 Sep;31(9):743-8.
Khera PK, Joiner CH, Carruthers A, Lindsell CJ, Smith EP, Franco RS, Holmes YR, Cohen RM. Evidence for interindividual heterogeneity in the glucose gradient across the human red blood cell membrane and its relationship to hemoglobin glycation. Diabetes. 2008 Sep;57(9):2445-52.

---

## Round 0.2 · accepted · Accept

The manuscript has been improved, by adequately addressing the issues raised, and suggestions proposed by the reviewers.

Reviewer 1 ·

Basic reporting

Authors have revised the manuscript and fulfilled most of my suggestions.

Experimental design

Authors have revised the manuscript and fulfilled most of my suggestions.

Validity of the findings

Authors have revised the manuscript and fulfilled most of my suggestions.

---

## Author Rebuttal · Round 0.2

PEER J Manuscript #2015:03:4505:0:0

Marked increase in rat red blood cell membrane protein glycosylation by one-month treatment with a cafeteria diet, by L. Oliva, C. Baron, X. Remesar, J.A. Fernández-López & M. Alemany

**Response To the Editor and Reviewer's comments/suggestions/requirements**

Editor:

Many methodological aspects should be addressed, including experiment replication, glycemic variability, normalization of the glycation plasma and RBC products. Also, the counterintuitive finding of blood cell glucose reduction following diet should be discussed.

Please be aware that we consider these revisions to be major, and your revised manuscript will probably have to be re-reviewed.

If you are willing to undertake these changes, please submit your revised manuscript (with any rebuttal information*) to the journal within 60 days.

Thanks for a thorough, extensive and detailed revision of our work. We are fully aware that our data question the common justification of parallel increased in the percentage of $Hb_{1AC}$ versus total Hb and hyperglycemia. But for many years we have observed that glucose in red blood cells' (RBC) compartment is seldom a correlate of plasma glucose. The term glycemia is thus ambiguous in its meaning, since it is applied equally to plasma, serum and whole blood glucose. For this reason we have taken extreme care in establishing the validity of the methodology used and to present the data --when possible-- in a quantitative way.

We are conscious that our results open more questions than they explain, but we believe that general assumptions, so cherished in clinical practice should be either justified or discarded, and the glycosylation of hemoglobin is one of these unexplained conundrums. We hope that the extensive changes introduced in the text and our additional explanations will approach us to the objective of publishing in Peer J.

Reviewer 1 (Anonymous)

Basic reporting

The article is clearly written. Only a few typing errors should be revised. E.g. in the abstract change "inn" for "in" and remove last bracket in the last sentence of the abstract. The article is properly put into context, with enough references. Most figures are relevant and well explained, although in my opinion I would remove Figure 1 since the information that it contains is redundant with that of table 1.

Thanks for your comments. We have reread several times-person the text and found the typos you mentioned and an additional couple of them, as well as other writing errors, which we corrected. We apologize for the carelessness this represents, we have no other explanation than the superposition of drafts and the persistence of residual material not adequately removed.

With respect to Table 1 and Figure 1, we believe that both should remain in the text. The Table presents concentration of plasma in non anesthetized rats, but not the blood levels (for technical reasons, no whole blood samples were used). Table 2 presents, however the levels of glucose measured in plasma and blood of anesthetized rats, the same used in Figure 1. However, what Figure 1 presents are not concentrations but the distribution of whole blood glucose between the cells and plasma compartments, an information that could not be directly extracted nor visualized from the concentrations alone.

Experimental design

The manuscript describes original research in an interesting field. The experimental design is appropriate. There are some concerns about the methodology, but the authors address it properly in the discussion. Methods are detailed, although in my opinion a brief description of the cafeteria diet used would be of interest. If metabolic cages were used to quantify ingested food it should be stated. Also, there is an allusion to unpublished results in the materials section (line 101) that is not clear what it refers to. Ethics issues have properly been addressed.

As requested, a description of the simplified cafeteria diet used, as well as a few clarifying details of the handling and maintenance of the animals have been now included in the text.

The reference to unpublished results, as the Reviewer noted, was unnecessary and perhaps misleading, it has been removed.

Validity of the findings

The design contemplates enough replicates and statistics have been addressed to rely on the data. I recommend adding statistics concerning the increase in body weight that is mentioned in the first paragraph of the results section. I also suggest removing lines 170-172 in which authors mention data not shown that is normalized in a way that they have just discarded as reliable enough. In the discussion section, the lines 227-229 should be better explained since the meaning is not clear. Conclusions are clearly stated.

As requested, statistical data on body weight change have been added to Table 1. The text on lines 170-172 has been changed to include the data omitted (in response to a query of Reviewer #2). Thus, the text reflects not only a reference of HMF liberated per unit of phospholipid OP but also with respect to membrane protein (Figure 3, now modified to include these data).

We have completely rewritten the paragraph including lines 227-229, we agree, it was unreadable. We hope the new redaction helps the reader to follow our line of thought.

Reviewer 2 (Anonymous)

Basic reporting

see below

Experimental design

see below

Validity of the findings

see below

Comments for the author

This manuscript is directed towards comparing differences by gender and by diet in relative glycosylation of plasma proteins, hemoglobin and red cell membrane components in rats as a means of assessing changes with an animal model of metabolic syndrome. The manuscript would come across as more focused if the background and objectives portion of the abstract contained objectives.

We intended to focus the study on the early period of diet-induced metabolic syndrome, and use this model to explore the question of glycosylation of hemoglobin presented above. We included male and female animals because of the wide range of metabolic differences existing because of sex on the manifestation of metabolic syndrome, but also on haemodynamics .

We agree with the Reviewer that the Abstract lacked to present clearly our objectives. We have now included a sentence summarizing our intention in carrying out this study.

I have a number of concerns about this manuscript. Essentially all the data are from a single experiment with no evidence the results were reproduced. This is probably most important with the estimate of marked reduction in blood cell glucose with diet, one of the more dramatic findings. The result doesn't make much sense physiologically, hasn't been reproduced, and depends on a subtraction methodology full of assumptions and for which there are considerable pitfalls. The membrane glycosylation is expressed per unit phospholipid phosphorus rather than per unit protein because the authors are concerned that they can't distinguish what protein is membrane and what protein is hemoglobin contaminant. However if that is a concern how can they depend on the fraction of the HMF generated being from membrane proteins and not from contaminating glycated hemoglobin?

We have found strange lacks of relationships between glycemia and glycosylated hemoglobin (a term that usually refers only to $Hb_{1AC}$). We had serious problems measuring something so simple as glycemia in humans and rats (hence the use of mutarrotase), and the constant confusion between blood, plasma and serum. There are also many papers by other Authors that question the validity of glycosylated hemoglobin correlations with mean glycemia. We devised a specific experiment, with a sufficient number of animals to get statistically significant results. This is the usual procedure for testing any hypothesis, and we did that.

We have used this same approach previously (Palou et al. Diabet Metab 1980, cited in the text) to present a mechanism that facilitates the transfer of glucose though the placenta from mother to foetus. The method used to calculate the glucose in the cell compartment has been widely used (including one of the references that the Reviewer kindly provided in his/her last query).

The question about cell glucose reduction with cafeteria diet not making sense from a physiological point of view is a serious question, which we initially shared. However, experience has shown us that when results are stubbornly showing something, Okham's razor reasoning point to what is correct are the data and not necessarily our interpretation. We searched extensively the literature, going back to the time when everybody now believes that "these things were established" and found that the assumed permeability of RBC membrane was not free but controlled by GLUT1 transporters. The data we found

(and a few are cited in the text) leave crystal clear that the assumption that plasma glucose and RBC glucose were in equilibrium at the same concentration (obviously in molal units) was not based on experimental evidence. For instance, we know that the RBC is glycolytic, using glucose to obtain ATP and release lactate; that means a rapid conversion of any free glucose to glucose-6P by hexokinase. This compound retains the aldehyde function on C1 but it is not glucose. So, a combination of control of both GLUT1 and hexokinase, both sensitive to insulin (in their expression during haematopoiesis), but affected by other plasma (or RBC) metabolites may deeply modify the distribution of glucose within the blood. We are now preparing an experiment to check the glycosylating ability of other monosaccharide and glycolytic intermediary metabolites on this point. We don't have an explanation yet, but the results are clear and don't support the currently accepted hypothesis.

The mention on the possibility of pitfalls and errors in the estimation of RBC intracellular glucose are valid (and essential for a manuscript of the type of the one we presented). We have taken a considerable amount of time and effort to know, control and minimize any sources of error. For instance, we measured blood and plasma glucose just in a few minutes after the animal was killed, and dit it using a method (perchloric acid) that broke all cells. We tried, previously to measure glucose in packed cells and found that it was almost always close to zero. RBC are living remnants of cells, with active glycolysis, and it is expectable that all glucose will be processed when not in contact with a constant supply (plasma). The question of trapped plasma has been already explained, addressed and published. In any case, only by looking at the blood and plasma glucose levels (Table 2) one can see that the concentration in plasma is higher, which means that the concentration in the cells must be lower. We have added a short paragraph in the Discussion presenting this fact as additional proof of concept. Our quantification of the differences may be discussed, and probably is not perfect, but as far as we know it is the only way we have found to try to solve this conundrum, and we have covered all bases except the obvious question of anaesthesia, that could not be avoided for ethical reasons which we obey and with which we fully agree.

The problem of contamination of membrane proteins with spectrin and hemoglobin was one of the problems more difficult to correct. We purified RBC membranes using an standard procedure, and in each cleaning pass we lost a sizeable amount of membrane. We decided to begin with more material and finish the process when there was no trace (by looking against a white paper) of pink. This was not proof that there was not hemoglobin in the samples, and worst, that there were no spectrin threads bound to the membrane, and for that reason we used the phospholipid P measurement. However, we have many data on this question and, following your question and a reference by Reviewer #1 we have now included the level of glycosylation of these purified membrane proteins in Figure 3. The pattern of glycosylation is similar in these proteins to that found when referred to phospholipid P, but the statistical significance was poor. In any case, there are clear differences between groups, which contrast with the considerable uniformity of both plasma and hemoglobin glycosylation presented in Figure 2. This is an indirect proof that membranes were adequately purified and that the contamination with non-membrane proteins was probably minimal, since otherwise, the parallelism with phospholipid-P would be hard to explain..

The ratios of glycation residue per hemoglobin and per albumin molecule do not sound plausible. It would be well for the authors to review relevant literature and see how consistent those findings are with previous reports. It doesn't sound like necessary positive and negative controls are reported with those determinations to support the validity of the quantitation.

We have modified the text of this part of the Discussion in order to clarify the point raised by Reviewer. #1 We did as the Reviewer suggested, and found that after $Hb_{1AC}$ was isolated and identified, the measurement of glycosylated hemoglobin (i.e. glycosyl residues bound over all free amino groups available) ceased to appear in the literature. Only very old papers (cited in the text) referred to glycosylated hemoglobin *stricto sensu*, that means that we have no previously published data in sufficient number to compare our data. We had to compare, instead, with the data for $Hb_{1AC}$, for which there is a considerable amount of data available. However, many of the extrapolations of the concentration of this modified hemoglobin to mean daily glycemia do not come from direct experimental data, since in all cases it was assumed that the RBC membrane was freely permeable to glucose and in equilibrium with plasma. In the Discussion, we have have added, now, that the degree of hemoglobin glycosylation we found was in the range of six-fold that corresponding to the terminal amino groups of $Hb_{1AC}$, The number of free amino groups in hemoglobin is much higher than that, so the data may be well within the expected range.

The authors do not adequately explain why there are such marked dIfferences in glucoses between tail vein blood and blood obtained by exsanguination. There are some comments about effects of the anesthetic to raise blood glucose but they are phrased in a fashion that is not transparent to a reader who was not a participant in the work. An inadequate explanation of those differences tends to cast doubt on the validity

of other findings obtained on those samples.

We have rewritten part of the justification of the differences. As we have explained above, to obtain sufficient blood to carry out the experiments we had to use a humane way to sacrifice the animals, in our University, isoflurane is the preselected method. But isoflurane raises considerably plasma glucose, a well known fat that we have reinforced now with more bibliography. We included the previous tail cut plasma glucose estimation as a way to present the data in all their extension, and to document a weak point of our study, the need for anaesthesia, a point that could not be circumvented. Our only response to the valid question posed by the Reviewer is the support of bibliography and the limitations we acknowledged by using the "basal state" plasma glucose level for comparison. We hope that the redaction is now more clear and conveys the facts we have repeated here.

The authors allude to differences in level of glycation per unit protein between plasma proteins and RBC proteins without conveying that they understand why those are and they express surprise at the greater extent of glycation of intra-RBC proteins than of plasma proteins. There are two key concepts commonly invoked for those findings which the authors don't convey awareness of: one is differences among animal species in the permeability of the RBC membrane to glucose and the other is the dependence of accumulation of glycated residues on the turnover time of the underlying proteins. Hb is present in the circulation for a much longer interval than albumin. There is also some evidence on correlations correlations between level of glycosylation in plasma vs in red blood cell proteins which may be of interest:

Higgins PJ, Garlick RL, Bunn HF. Glycosylated hemoglobin in human and animal red cells. Role of glucose permeability. Diabetes. 1982 Sep;31(9):743-8.

Khera PK, Joiner CH, Carruthers A, Lindsell CJ, Smith EP, Franco RS, Holmes YR, Cohen RM. Evidence for interindividual heterogeneity in the glucose gradient across the human red blood cell membrane and its relationship to hemoglobin glycation. Diabetes. 2008 Sep;57(9):2445-52.

Thanks for the references, we have included them in the text. The Khera paper presents a hyperbolic curve of glucose permeability that clearly shows the existence of a transporter (in fact facilitated transporter) for glucose permeation into the RBC. The Higgins paper adds strength to the wide range of variation between species (i.e. pigs seem to have no glucose in RBC), individuals but also modulation of permeation. This is our point, and this, alone, may help explain most of the differences observed. The question of the transporter was already solved almost half a century before us, but the data remain valid and reliable. In fact they help us explain the results.

A question, basic in Physiology, is the trend to preserve functions in similar ways for animals of a similar lineage. It is not reasonable to assume that glycolytic extra-simple cell remnants such as RBCs would show a widely different behavior between related mammal species, that is a reason why we use rats or pigs in our experiments, apart from questions of scale, allometric correlations, metabolic paths are similar, regulation fairly similar and the main differences are die to size. These factors alter cell turnover, but not glycolysis, they may affect the velocity of the transport, but not the non-enzymatic glycation of proteins, depending on substrate concentrations and temperature. In this sense, we agree with the Reviewer that hemoglobin stays longer in blood than albumin, since their turnover rates are in the 40x to 80x range for a rat. But there are two critical factors that also influence the results: a) the number of amino groups where a Maillard adduct can be formed by the free aldose (when most of them are glycosylated, no further glycosylation is possible), and b) the fact, the stubborn fact, that hemoglobin is not in direct contact with plasma glucose. This has been all the time our thesis, and we believe we have presented sufficient references and original experimental data to prove this fact. Hemoglobin is packed tightly within the RBC, not free in the plasma as is the case with albumin, this can explain the differences. Our surprise stems from the scant amount of glucose (free glucose) available within the RBC. We are working on the hypothesis presented above to understand this difference.

Thanks